# “We’re Not Going to Be as Prepared”: A Qualitative Study of Healthcare Trainees’ Experiences after One Year of the COVID-19 Pandemic

**DOI:** 10.3390/ijerph20054255

**Published:** 2023-02-27

**Authors:** Holly Blake, Alex Brewer, Niki Chouliara

**Affiliations:** 1School of Health Sciences, University of Nottingham, Nottingham NG7 2HA, UK; 2NIHR Nottingham Biomedical Research Centre, Nottingham NG7 2UH, UK; 3School of Medicine, University of Nottingham, Nottingham NG7 2UK, UK; 4NIHR Applied Research Collaboration East Midlands, Nottingham NG7 2TU, UK

**Keywords:** COVID-19, higher education, healthcare, students, mental health, qualitative

## Abstract

The COVID-19 pandemic had significant impacts on the mental health and academic experiences of healthcare trainees. Building on findings from earlier in the pandemic, we explore the impacts on healthcare trainees after a sustained pandemic period of 12–14 months, involving multiple lockdowns, changes in government COVID-19 regulations and the delivery of health education. A qualitative study was conducted between March–May 2021. Participants were 12 healthcare trainees (10 women, 2 men) of medicine, nursing, and midwifery, registered at one of three higher education institutions in the United Kingdom. Interviews were fully transcribed, and data were thematically analysed using a combination of deductive and inductive approaches. We identified three key themes with eight subthemes: (i) academic experiences (adjustment to online learning, loss of clinical experience, confidence in the university), (ii) impacts on wellbeing (psychosocial impacts, physical impacts, pandemic duration and multiple lockdowns), and (iii) support frameworks (university preparedness for increased student support needs, importance of relationship with academic tutors). Findings shed light on the long-lasting and emerging impacts of the pandemic over time. We identify support needs for trainees both during their academic studies, and as they move forwards into professional roles within the healthcare workforce. Recommendations are made for higher education institutions and healthcare employers.

## 1. Introduction

The coronavirus (COVID-19) pandemic has negatively impacted the mental wellbeing of the general population worldwide [1,2,3,4]. Globally, there has been high rates of anxiety, depression, post-traumatic stress disorder, psychological distress and stress [1,2]. In 2020, anxiety and depression were found to be highest in people living with pre-existing conditions and those infected with COVID-19 [3]. Common risk factors associated with mental distress during the COVID-19 pandemic in the general population include female gender, younger age group (≤40 years), presence of chronic/psychiatric illnesses, unemployment, student status, and frequent exposure to social media/news concerning COVID-19 [2]. Healthcare workers (HCWs) are at the forefront of the pandemic response and have experienced significant psychological impacts of COVID-19 [5,6,7,8]. The ICON study explored the impacts of COVID-19 on the UK nursing and midwifery workforce across three time points in 2020 and found that nurses and midwives experienced a high prevalence of negative psychological effects, including severe stress, severe anxiety, and signs of post-traumatic stress disorder [5]. Similarly, a systematic review and meta-analysis focused on doctors, nurses and allied health professionals found that post-traumatic stress disorder was the most common mental health disorder associated with the COVID-19 pandemic among health care workers, followed by anxiety, depression, and distress [7]. There has been a high prevalence of insomnia and burnout in HCWs during the pandemic [6]. Increased psychological distress in HCWs has been associated with personal factors, such as younger age and caring responsibilities, and workplace factors such as a lack of personal protective equipment (PPE) and lack of access to, or confidence in, essential training [5]. It has been suggested that the mental health impacts of the pandemic on HCWs may be higher in certain occupational groups (e.g., health technicians, medical students, and frontline health workers) [8] and, overall, may be underestimated [6].

As the future healthcare workforce, trainees are capable and willing to be involved in global health emergencies [9], although the mental health of healthcare trainees has also been impacted during the COVID-19 pandemic [10,11,12,13,14,15,16]. Systematic reviews and meta-analyses found that student nurses reported suffering from fear, stress, anxiety, depression, and sleep disturbance [10,12]. Similarly, rates of stress, anxiety and depression are high in medical trainees [13,14,15,16], and some studies identified evidence of suicidal ideation and burnout [17]. The mental health impacts of the pandemic have influenced healthcare trainees’ intentions to leave their training [11]. This is not unexpected since systematic reviews show that students and younger age groups are particularly at risk for pandemic-related distress [2,17], and college/university students more broadly have experienced high rates of anxiety and depression during the pandemic [18,19,20,21,22], often associated with social restrictions and periods of self-isolation [23,24,25]. However, the prevalence of depression and anxiety during COVID-19 is relatively higher among healthcare trainees than both the general population and healthcare workers (e.g., [8,13]).

A study conducted in the first few months of the outbreak of COVID-19 in the UK, highlighted the impacts of the pandemic experienced by healthcare trainees studying medicine, nursing, midwifery and other healthcare disciplines [26]. Among myriad challenges were the disruption to academic studies, rapid transition to online learning, social isolation, mental health impacts and challenges to accessing mental health support. Importantly, trainees in this study raised concerns about the future in terms of the negative impact of the pandemic on their education, and whether resulting gaps in their knowledge would leave them unprepared for their future clinical practice. These were early experiences, 4–5 months after the pandemic outbreak in 2020, in the context of high fear associated with a new and rapidly escalating virus, and higher education institutions operating in ‘crisis mode’ [27], rapidly implementing virus containment and mitigation strategies, and overhauling systems and processes for the delivery of teaching and learning. Many of the reported reviews also draw on evidence from earlier in the pandemic. Therefore, in the current study, we build on early findings to explore the impact of the COVID-19 pandemic on healthcare trainees after a sustained pandemic period of 12–14 months. Findings will shed light on any long-lasting or emerging impacts of the pandemic and identify support needs for trainees going forwards.

## 2. Methods

### 2.1. Study Design

This was a qualitative interview study, conducted as part of the larger PoWerS research programme [26] which had explored the impacts of COVID-19 on healthcare students six months after a pandemic was declared in the UK. Here, we explore the impact of the COVID-19 pandemic on healthcare students in the UK after a sustained pandemic period, specifically 12–14 months on. This follows multiple lockdowns and changes in government regulations relating to COVID-19 and social restrictions, alongside an extended period for higher education institutions to adjust approaches to the delivery of healthcare education and student support for learning. The consolidated criteria for reporting qualitative research (COREQ-32) checklist [28] was used to ensure the quality of reporting this study (Appendix A). Ethical approval for the study was obtained from the University of Nottingham Faculty of Medicine and Health Sciences Research Ethics Committee in March 2021 (FMHS REC ref 39-0620).

### 2.2. Participants and Setting

Participants were medicine, nursing and midwifery trainees registered at higher education institutions in the United Kingdom (UK). Trainees from other disciplines and those who were not registered for study during this period were excluded. Participants were recruited via social media and email promotion of a study advertisement via student society circulation lists.

### 2.3. Procedure

Qualitative data were collected over a 7-week period between March and May 2021. In response to the study promotion, potential participants were asked to contact the researcher by email to express interest in taking part. They were emailed a participant information sheet that explained the purpose of the study, the research processes and an invitation to take part in a single interview to share their views. All participants provided written informed consent before the interview took place. To optimise recruitment during a challenging pandemic period, trainees were given the option of entering a prize draw to win a 25GBP shopping voucher which has been shown to increase response rates in research [29]. No trainees refused to participate after expressing interest or dropped out after consenting to take part. Recruitment continued until the dataset was deemed to hold sufficient information power [30]. Semi-structured interviews were conducted online using Microsoft Teams and were audio-recorded with consent. Interviews were fully transcribed using an online transcription software and then checked by the researcher to ensure their accuracy. Due to time constraints, the transcripts were not returned to the participants for comment and/or correction. An interview topic guide (Appendix A) was generated to provide a foundation for the interviews. The topic guide was informed by the PoWerS study [26] and was finalised through discussion between the research team and a healthcare trainee who was not a participant in the study to agree on the relevance of the questioning. Our approach and interpretation were informed by published recommendations for virtual qualitative health research conducted during a pandemic [31].

### 2.4. Data Analysis

The data analysis process used in the current study employed thematic analysis [32] using a combination of deductive and inductive approaches. Initially, a coding framework was developed based on key areas identified in a previous study [26]. The key areas of discussion were as follows: impacts on wellbeing, impacts on academic studies and learning, and support for healthcare students. Under each overarching category, codes were generated that were grounded in the interview data. One researcher (AB) coded the data, and a second researcher (NC) checked and verified the codes. We did not code data that did not fall under these categories and was not relevant to our research questions. The initial overarching framework was then refined based on the data and generated codes. In accordance with the process of thematic analysis, a process of abstraction then took place whereby codes were grouped together into subthemes and then into main, overarching themes.

## 3. Results

In total, 12 participants provided informed consent and were interviewed (10 women, 2 men) registered at one of three universities in the UK (site 1: *n* = 9; site 2: *n* = 2, site 3, *n* = 1). Interviews lasted between 20 and 37 min, with an average time of 33 min. Participant age ranged from 19–42 years. Nine had worked in the UK health or social care environments in areas considered high-risk for COVID-19 during the pandemic (e.g., as defined in [26]; this included dedicated COVID + ve ward, intensive care unit, emergency department or ambulance services, ward with COVID + ve patients, entrance meet and greet, staff or regular visitor to care or residential home, or other self–defined high–risk area). Participants were trainees of medicine (*n =* 7), nursing (*n* = 2) and midwifery (*n* = 3). Table 1 shows participant characteristics.

Drawing on the results of the thematic analysis of the qualitative data, three main themes and eight sub-themes were generated (Figure 1).

### 3.1. Theme 1: Academic Experiences

#### 3.1.1. Adjustment to Online Learning

Trainees had diverse experiences and opinions relating to the sudden transition to online learning due to social restrictions during the early stages of the pandemic. The lack of social interaction with online learning was challenging for some who had: *“gone from everyday being in-person and being really interactive to just sitting in your room”* (P5). Some trainees raised concerns about the increased use of pre-recorded lecture materials, particularly in the early pandemic days, referring to: *“very poor-quality recordings of lectures from previous years, some of which were completely incomprehensible”* (P2). For others, pre-recorded sessions provided an opportunity for study without interruption and offered greater flexibility for trainees, which was valued. Live online lectures were viewed more positively than pre-recorded materials as they allowed trainees the opportunity for interaction and conversation, either verbally or via chat functions within the video-conferencing platform. Despite the challenges experienced by trainees over the course of a year, there was a clear recognition of the workload and challenges involved for educators in diverting traditional approaches to teaching and learning to online platforms at pace and scale. Trainees spoke positively of the immense efforts and achievements of academic staff in continuing to deliver higher education through such unprecedented circumstances: *“…apart from the odd technical blip at the beginning, it’s been pretty seamless. I mean, from, from people that have not done this before to pull off what they pulled off has been amazing. I can’t fault them at all.”* (P7).

#### 3.1.2. Loss of Clinical Experience

Many trainees felt they had lost essential clinical experience during the pandemic: “*It felt like taking a step backwards. Having all the course moved online…that’s what we were doing in pre-clinical years… suddenly, we’re back in front of the computer and it was quite demoralising*” (P2). The impact on clinical experience varied according to the level of study and degree programme. Those in their earlier stages of study were less concerned about the impact of the pandemic on their clinical learning. Due to the strong emphasis on theory and less time in clinical practice in year one, students lost little or no practice experience and were confident they would have the necessary experience by the time they graduated. For those in the later stages of study, the pandemic had a greater impact on clinical learning as they were more likely to have missed clinical placement time due to lockdowns or social restrictions. A second-year medical student reported they “*haven’t really had clinical placements this year*” (P4); a fourth-year medical trainee reported having “*missed a good six months of placement*” at the start of the pandemic (P2). This led to worry about the impact of missed or altered clinical exposure on academic learning and performance: “…*our clinical opportunities are greatly reduced because of the pandemic...we’ve definitely had a huge impact on our education*” (P2). Since the pandemic was long-lasting, this worry continued even after social restrictions were relaxed and placements had been reintroduced. Some trainees reported that clinical experiences during the pandemic were not of the same quality, whereas others felt well supported on placements (by clinical mentors and placement coordinators) and valued the learning experiences these opportunities provided. Views towards clinical learning on placements varied according to where trainees were placed (e.g., depending on the hospital or department, or level of mentor support). Some trainees had signed up to aid healthcare organisations in various roles (e.g., healthcare assistant, logistics) in which they gained further exposure to clinical environments and felt a sense of contribution to the pandemic response.

Trainees expressed concerns about preparedness for future clinical job roles, with some speaking of “*imposter syndrome*” (P2) as they felt they may not graduate at the same professional standard as others and would be unworthy of their professional title: *“something we’ve had to come to terms with that we’re not going to be as prepared”* (P6). Trainees highlighted a need for clinical mentors to recognise the impact of the pandemic on clinical learning when students transition to professional posts (e.g., lost opportunities to practice clinical skills in clinical environments; associated loss of confidence), and a desire for future supervisors to recognise that they may need more support than their predecessors: “*hope that our seniors are understanding and… are able to guide us when we get to that stage”* (P6). Although there were concerns about preparedness for future practice, and an impact on confidence levels, trainees did not equate this with an impact on their future employability. Irrespective of their perceived amount of (and confidence in) clinical experience, it was perceived that: *“doctors are going to be needed anyway”* (P4).

#### 3.1.3. Confidence in the University

Participants had mixed views relating to their confidence in the university’s approach during the pandemic to the conduct of assessments and preparing trainees for clinical practice. Due to COVID-19 restrictions, several participants had some, or all, of their practical assessments changed to online format. Some alluded to inequity in the experience of online assessments; one trainee referred to the challenges of finding an appropriate environment in which to undertake online examinations: *“[I] didn’t have a quiet place in my house to do the exam…family was going in and out of the house and being somewhat noisy”* (P3). Some participants raised concerns about the lack of standardisation in the university’s approach to conducting and monitoring take-home examinations and the potential for cheating, particularly among medical students: *“I heard a lot of rumours that people were doing the exams together… and some of my friends had suspiciously high marks compared to what they usually get relative to me. So, I think that was very upsetting.”* (P3)*; “there wasn’t any sort of system put in place to stop people cheating”* (P6). Other trainees offered a more positive perspective on their university’s approach, both to the assessment of learning and preparation for future clinical practice. They believed that the challenges of the pandemic would be taken into consideration in grading. In relation to clinical examinations, one medical trainee reported: “*I haven’t had much clinical experience, but then again, I think the uni did tell the examiners to take it into account. And I do think everyone’s in the same boat so, you know, the marks will naturally shift.”* (P8). Some trainees suggested that the university was likely to generate opportunities to recoup any lost clinical learning time.

### 3.2. Impacts on Wellbeing

#### 3.2.1. Psychosocial Impacts of the Pandemic

Most of the trainees commented on the negative impacts of the pandemic on their mental wellbeing. After more than a year of lockdowns, social restrictions (e.g., social distancing, self-isolation), and studying from home, several students felt this had impacted on their ability to live and study as they had done previously. Some trainees spoke of having to take time away from studies or placements due to the impacts of COVID-19 on their mental wellbeing: *“…some time off placement to make up at the end of the course”* (P10).

Several of the trainees experienced anxiety that was directly associated with the pandemic; for some, this was related to concerns about catching and transmitting the virus (e.g., to peers, colleagues, or vulnerable patients). This was discussed either in relation to their potential for contact with COVID-19 patients on placements or in the context of social mixing. In terms of socialising, for some, the virus transmission risk outweighed the desire to socialise. For others, their anxiety centred around a perceived loss of social skills and the return to interacting with others after long periods of isolation. One midwifery trainee referred to feeling *“slightly agoraphobic…it’s just got worse the longer it’s gone on”* (P7). For this trainee, the negative impact of the pandemic on their mental wellbeing was profound and led to them considering terminating their studies: “*(the pandemic) has made me feel incredibly low… at times not knowing if I can carry on my course”* (P7). Not all students were impacted to the same extent. Those who considered themselves to be less outgoing and sociable by nature, reported being less impacted by remote working and social restrictions during the pandemic than those who would usually socially interact to a greater extent: *“I’m quite introverted anyway, so it (isolation) has not had a massive psychological impact on me”* (P1). Other trainees proposed that their mental health during the pandemic could be improved through increased opportunities to socialise, even if this needed to be online (e.g., “pen pals” or evening social events).

Some trainees experienced a drop in their motivation for study during the pandemic, which worsened as the pandemic went on, with trainees: “*starting to struggle to maintain any kind of momentum, finding the will to do it”* (P7). Low motivation was associated with poor mental wellbeing, a loss of social interaction with peers and removal of in-person contact with academics during lockdowns: “*no-one’s really keeping you accountable”* (P3).

#### 3.2.2. Physical Impacts of the Pandemic

Although participants spoke more frequently about the psychosocial impacts of the pandemic, several of the trainees discussed the impacts of COVID-19 on their physical health. Impacts could be positive or negative depending on the individual, their circumstances and prior lifestyle behaviours. With the increased time spent at home to study, some trainees were consuming more unhealthy foods, whereas others reported they had less access to ‘junk food’ and their diet had improved: “…*when you’re going about doing everything, you just have a pack of crisps here, a cake there, whatever. But when you’re just sat about the house, you can feel it clogging up your arteries”* (P9). Similarly, some had exercised more, and others less, compared with pre-pandemic behaviours. The closure of gyms and running tracks had impacted those who had previously been very physically active. Two participants reported experiencing sleep issues during periods of pandemic-related lockdown or social restrictions, and this was associated with working either in environments that were not fit for study or balancing work and home life: *“very hard … sleeping and doing everything in the same environment”* (P4).

#### 3.2.3. Pandemic Duration and Multiple Lockdowns

At the point of interviews, the COVID-19 pandemic had continued for over a year, with key lockdowns and social restrictions introduced between March 2020 and May 2021. Several participants commented on how the impact of these restrictions varied across time. This was attributed to the progression of the pandemic, the weather, the timing (whether lockdowns occurred within or outside of academic term-time) and the presence (or lack) of social support through the year from peers, friends, and family. The wellbeing impact of the first lockdown had been less significant for some, as it occurred during a period of warm weather (spring/summer), and trainees had returned to family homes and were therefore accessing social support: *“… because I’ve got a lot of siblings the house it was still quite busy and it kind of felt like normal life to be honest”* (P3); *“… just being home for the lockdown was just nice because they’re my support system and they’re always there when I need them.”* (P4). Some trainees raised that returning to family homes during lockdowns was a protective factor, providing a sense of safety and security in a rapidly changing global context. However, for those with young families, balancing childcare alongside studies was a particular challenge, especially during lockdowns and periods of self-isolation. With exceptions, as time went on, the pandemic took a heavy toll and wellbeing declined over time for most: *“the duration of the lockdowns has definitely affected my mental wellbeing.”* (P12); “*…the thought of working from home and actually the pace being slower was actually quite nice because the course moves at a hundred miles an hour… but it quickly became apparent actually that it wasn’t a gift”* (P7).

### 3.3. Support Frameworks

#### 3.3.1. University Preparedness for Increased Student Support Needs

There was a consensus that structured mental health support services offered by universities during the pandemic were poorly communicated (i.e., in terms of what was available and how it could be accessed) and inadequate. Perceived problems with structured services seemed to be associated with access rather than the quality of provision. A minority of those raising psychological impacts of the pandemic did not know where or how to access help and support. A few trainees experiencing difficulties had reached out to counselling services. While those who had accessed counselling generally reported a good experience, several indicated that support was often not timely due to long delays in accessing appointments. Some trainees referred to specific efforts being made to improve wellbeing within their academic faculties, for example, course leaders providing protected time-out from study, and advocating attendance at structured wellbeing sessions. However, it was viewed that there was a focus on quantity of wellbeing sessions, rather than quality. Some trainees felt that the impact of wellbeing provisions on trainees was not fully considered, as trainee workloads were not reduced to allow attendance at wellbeing sessions. Therefore, academic activity was compressed into a shorter timescale to compensate for this, resulting in an increased work intensity, and paradoxically, negatively impacting wellbeing. While experiences were broadly comparable across trainees from different institutions, disciplines and years of study, the lack of clarity around mental wellbeing support provisions seemed to be particularly notable for medical trainees: *“If I was going through something, I wouldn’t know who to contact”* (P5).

#### 3.3.2. Importance of Relationship with Academic Tutors

While structured support services and online mental health support were viewed more negatively, most trainees spoke about the value of regular contact with academics, particularly personal tutors, in ensuring they had a positive learning experience and supporting their wellbeing throughout the pandemic. Student representatives augmented support from personal tutors, and trainees applauded their efforts to support the communication of information about course-related changes and welfare support at critical times during the pandemic. In contrast, some trainees had reached out to academics for support, but were dissatisfied with the response and felt that support needs were not being met: *“…when we’ve raised concerns as a cohort, about, you know, our deteriorating mental health as a group… A couple of the lecturers didn’t seem to be quite so sensitive to that. It was very much “well we’re all in the same boat, everybody’s struggling”, that was difficult to swallow.”* (P7). Trainees’ experiences varied depending on their relationship with their personal tutor and the level of support that individual was willing or able to provide. Trainees who reported a good relationship with their personal tutor highlighted the support they provided and the positive impact of this on their mental wellbeing, particularly during lockdowns. In general, trainees placed a high value on tutors who actively reached out to provide support, rather than waiting passively for trainees to contact them with issues: *“he just checks in, makes sure I’m okay… he’ll ask if I have any concerns”* (P5); “*I’ve got an amazing personal tutor who’s really supportive and really responsive”* (P3).

## 4. Discussion

This study explores the impact of the COVID-19 pandemic after a sustained pandemic period of 12–14 months. Three key themes were generated from the data: (i) academic experiences, (ii) impacts on wellbeing and (iii) support frameworks.

In terms of academic experience, as identified in prior research [26], the rapid transition to online learning had been a significant stressor for healthcare trainees, although this was primarily related to the loss of social interaction with peers and tutors and was more problematic earlier in the pandemic and during periods of lockdown. Healthcare students have needed to adapt to a rapid transition to the use of technology to deliver education remotely and enable the continuation of teaching through periods of lockdown and social restrictions [33]. According to trainees in our sample, there was a difference in the value attributed to online teaching according to the nature of its delivery; live online lectures were generally perceived more positively than pre-recorded materials (particularly low-quality recordings). Our trainees felt that live online delivery allows interactivity and active collaborative learning, which is known to enhance critical thinking [34] and active learning [35] in a higher education context. While virtual teaching is purported to be effective, work is needed to further enhance student engagement and interactivity [36] and as noted by Rudolph and colleagues, “technology should not be revered as a panacea” [37]. With online learning likely to constitute at least a proportion of higher education teaching in the future, it seems advisable to ensure adequate attention is paid to strategies that foster online social interactions between faculty staff, clinical mentors, and healthcare trainees.

There were some aspects of taught course delivery that were viewed to be better delivered in-person. Specifically, the experience of technology-delivered remote examinations was not positive, in our sample. Remote examinations exposed inequalities in student experience during this time, largely due to the differences in home environments as some trainees did not have access to suitable environments to sit examinations, which they felt affected their performance. Social and digital inequalities have been noted in other student samples. For example, Bashir and colleagues [38] found that 61% of biosciences students at a UK institution were able to study uninterruptedly during an online examination period, with students (particularly those from more deprived households) reporting inadequate home working space/environment and lacking necessary items such as a desk [38]. Moving forwards, higher education institutions should consider exploring whether all trainees have access to reliable and affordable physical devices (e.g., computers, laptops), Internet connectivity and quiet study spaces in which to take online assessments—and ensure available institutional facilities are well-promoted to all trainee cohorts. While some trainees adopted a “*we’re all in the same boat”* attitude and believed the university would take the uniqueness of the pandemic situation into consideration with regards grading, other trainees were concerned about quality standards during this time and the lack of fairness in remote assessment processes. These concerns impacted their confidence in the university to deliver and monitor online assessments fairly. This has been found in other higher education settings, and participants have raised concerns relating to security, validity and fairness in the implementation of online assessments [39]. To mitigate these challenges, Shraim [39] recommended that online assessments may be best utilised for formative rather than summative assessments. However, there may be times when online summative assessments are necessary for trainees’ progression (e.g., for distance learning professional development courses or during future pandemics). Further work is therefore needed to ensure equity in implementation and quality standards, instill confidence in student cohorts relating to the fairness of the system.

Having had time to reflect on the year, trainees in our study highlighted the efforts made by academic staff to continue the delivery of teaching and learning during the COVID-19 pandemic. This is an important observation. COVID-19 has impacted higher education institutions around the world [40]. Adjusting to online learning has been as challenging for academic staff as for students [25,41]; academics have needed to overcome gaps in digital skills and reconfigure their pedagogical approaches to the online learning environment [41]. The challenges for academic staff have been immense, including unfamiliarity with the learning management systems, privacy concerns, issues with student engagement, increased preparation time and technological issues [42]. Over time, educators have made progress in transitioning from emergency measures to more pedagogically consistent approaches, albeit there remains a need for better integration of theoretical and practical learning [43]. Our study highlights that trainees are aware of the efforts made by academic staff to keep higher education functioning during this crisis. With a greater focus on students compared to academic staff in the published literature, we advocate that wellbeing and technological/pedagogical support would benefit staff engaged in the delivery of remote, online or hybrid education at all times (not least during a pandemic), coupled with monitoring of workload and stress levels as proposed elsewhere [25]. Supporting staff is an essential part of ensuring the provision of adequate and equitable support for healthcare trainees, and indeed, all students in higher education settings.

A key concern for healthcare trainees was the loss of clinical experience during the long-lasting pandemic, resulting in missed placements or lower-quality placement experiences. This was primarily an issue for those in later years of study, as found in previous research [26]. A lack of preparedness for future practice has been identified in other samples of nursing [44,45] and medical trainees [46]. Interestingly, trainees in our sample did not view this loss of clinical experience as a risk to their employability. However, trainees reported experiencing “imposter syndrome” due to a belief that their knowledge and skills would be lacking compared to predecessors. Imposter syndrome is not a new concept in healthcare and has previously been identified in both nursing [47] and medical [48] trainees. Since imposter syndrome is evident at every stage of the career [49] and is linked to burnout, anxiety, and depression [50], higher education organisations may consider addressing imposter syndrome as part of the preparation for the transition to professional practice, through workshops and training (e.g., [49]). Further, healthcare organisations and line managers should be mindful of the COVID-19 impacts on new recruits’ confidence to practice and ensure additional mentoring and training is in place to build confidence and address any perceived gaps in knowledge or skills in the initial employment period.

The impact of the pandemic and the shift to remote learning impacted on engagement in physical health behaviours for some of the trainees, including dietary and/or exercise habits and sleep patterns. With some exceptions, our trainees reported primarily negative impacts, which aligns with other studies showing that the COVID-19 pandemic impacted negatively on university students’ dietary intake, physical activity, sedentary behaviour, and sleep [51,52]. Advocating health behaviours is important for health status across the life course, since diet, physical activity and sleep are independently associated with health-risk indicators and all-cause mortality [53,54] and negative lifestyle behaviours are associated with lower psychological wellbeing [55,56]. With direct relevance to healthcare trainees, the prior finding that poor lifestyle choices in healthcare professionals or trainees can influence the likelihood of them role modelling health behaviours to others (e.g., colleagues, students), their views and actions towards promoting health to patients, and the willingness of patients to heed their advice [57,58,59,60]. Promoting the value of healthy lifestyles to health trainees (and health professionals) themselves during and beyond the pandemic, remains an important priority for health educators and healthcare employers alike. This will support future engagement with health promotion in patient care, and importantly, establish a healthy future workforce for health and care organisations.

Most notable were the psychological impacts of the pandemic on healthcare trainees, with participants in our study experiencing anxiety, low motivation for study, and low mood. Additionally, our trainees felt a sense of social responsibility as the future healthcare workforce; they feared contracting and transmitting the virus to vulnerable others. Fear of contracting the virus is high among college/university students more generally, irrespective of their subject discipline [61,62]. Our findings align with other studies that identified mental health impacts of the COVID-19 pandemic in the general population [1,2,3,4], healthcare professional [5,6,7,8] and healthcare trainee samples [10,11,12,13,14,15,16]. For some, the negative impacts of the pandemic on mental wellbeing were mitigated by environmental factors (e.g., good weather lifting mood), social support (e.g., from family members at home), or personality traits (e.g., those who reported introversion traits perceived reduced social contact to be less problematic). Conversely, other participants experienced significant wellbeing impacts that involved time out from studies and were sustained or increased over the duration of the year.

Our study shows that mental health impacts are still evident 12–14 months after COVID-19 was declared a pandemic. Using a conceptual model (Appendix A), a prior study presented relevant actions for mitigating the impacts of the pandemic on health and care workers; this showed to have relevance for healthcare trainees from diverse health and medical disciplines [26]. The main areas for action described within this model include proactive organisation approaches; psychologically supportive teams; communication strategies; managing emotions; social support and self-care. However, our study findings suggest that increased investment in mental health support may be required in the long term and not just as a short-term pandemic response. Mental health impacts of the pandemic were exacerbated by ongoing challenges in accessing supportive services. One year into the pandemic, some healthcare trainees still did not know where to access information about help and support for mental wellbeing, and others were struggling to access counselling services due to long waiting lists. Poor communication from their institutions relating to mental health support and inadequate capacity of student support services were issues identified by healthcare trainees in the first few months of the pandemic (e.g., [26]), but our study suggests this situation has not greatly improved. Given the continuation of the pandemic well beyond the current study data collection period, and the ongoing mental health impacts of COVID-19 for healthcare trainees, higher education organisations need to urgently invest additional resources into structured mental health support services. More attention is needed to raising awareness of mental health, and signposting to support, particularly in disciplines where mental ill-health can be considered taboo [63], and in student communities in which access to structured services for mental health is known to be low [23].

Outside of structured services, academics play a key role in student support. We observed that proactive approaches to trainee support were valued, particularly check-ins from academic tutors. We advocate that academic tutors are mindful of the sustained impacts of the pandemic on trainee cohorts and trainees’ concerns for their future as clinical practitioners. Tutors should be aware of the importance of their role in providing support and signposting to trainees, who may have increased support needs due to the long-lasting pandemic and its aftermath. This may require additional training and support for academics related to the role of the personal tutor. This would increase the parity of support that is provided across cohorts and subject disciplines and ensure that signposting to supportive services is both appropriate and timely. It has been argued that a combination of academic and mental health support is needed for healthcare trainees during a pandemic [64] and, of course, beyond. Recommendations for the study are summarised in Box 1.

Box 1Ten recommendations for higher education institutions and healthcare employers.Work is needed to further enhance student engagement and interactivity in online learning.Adequate attention should be paid to strategies that foster online social interac-tions between faculty staff, clinical mentors, and healthcare trainees.Higher education institutions should consider the impact of inequalities when remote assessments are conducted, e.g., ensuring access to quiet study spaces, reliable and affordable devices, and Internet connectivity.Technological/pedagogical support may benefit staff engaged in the delivery of remote, online or hybrid education, coupled with workload and stress monitoring, and wellbeing support.Promoting the value of healthy lifestyles to health trainees (and health profes-sionals) during and beyond the pandemic remains an important priority.Higher education organisations need to urgently invest additional resources into structured mental health support services to widen access and reduce waiting times.More attention is needed to raising awareness of mental health, and signposting to support, particularly in disciplines where mental ill-health can be considered taboo and in student communities in which access to structured services for mental health is known to be low.Proactive support from academic tutors, such as regular check-ins, will be bene-ficial given the sustained impacts of the pandemic on trainee cohorts. This may require additional training and support for academics related to the role of the personal tutor.Higher education organisations may consider addressing imposter syndrome as part of the preparation for the transition to professional practice through workshops and training.Healthcare organisations and line managers should be mindful of the COVID-19 impacts on new recruits’ confidence to practice, and ensure additional mentoring and training is in place to build confidence and address any perceived gaps in knowledge or skills in the initial employment period.

### Study Strengths and Limitations

Our findings are based on the views of a small sample of healthcare trainees from three healthcare disciplines. Although we gained insights across subject disciplines, our data did not allow us to explore similarities or differences between participants registered at different academic institutions. There were more female than male participants in our study (no trainees identified as non-binary), and so the views of male and non-binary trainees warrant further exploration. However, the gender imbalance in our sample broadly reflects trends in student cohorts and professional registrants from the disciplines included. For example, there is a higher proportion of female than male entrants, both to higher education in the UK [65] and to medical schools [66]. Further, the UK Nursing and Midwifery Council data shows that 89.3% of all nursing and midwifery registrants are female [67]. Our findings align with gender-based research, as participants in our sample reported beliefs aligned with “imposter syndrome”, which is more common in female than male healthcare trainees [68].

There may be a risk of selection bias in the study since trainees who were impacted more, or less, may have been more, or less, likely to take part in the research. For example, trainees who had experienced greater impacts from the pandemic may have been more likely to agree to take part in the study. Conversely, however, trainees struggling with mental health concerns may have felt less able to engage in research. Recruitment was challenging during the COVID-19 pandemic. This was not surprising given that our study highlights the impact of the pandemic on healthcare trainee mental wellbeing, which may have impacted on research engagement. In addition, the challenge of virtual recruitment into research studies during the COVID-19 pandemic has been recognised [69]. In our study, recruitment may have been facilitated by traditional, in-person recruitment efforts. However, the pandemic shifted research recruitment approaches to fully remote online platforms. Here, we focused primarily on the use of social media, although this has been identified as a valuable strategy for online recruitment to qualitative research studies [70]. We added a prize-draw incentive to encourage participation, which has been identified as a useful mechanism to increase uptake in research studies [70] and in this instance, helped us to achieve sufficient information power [30].

Future research may benefit from exploring the impacts of the pandemic on other healthcare disciplines. Further, there is a paucity of longitudinal research in this area and future studies might seek to explore whether there are changes in support needs over time, and the experiences of healthcare trainees who studied during the pandemic, as they transition into employment within healthcare organisations.

## 5. Conclusions

Building on findings from earlier in the pandemic, we conducted qualitative interviews to explore the impacts on healthcare trainees after a sustained pandemic period, involving multiple lockdowns, changes in government COVID-19 regulations and the delivery of health education. The COVID-19 pandemic had significant impacts on the wellbeing and academic experiences of healthcare trainees. The long-lasting duration of the pandemic has taken its toll; trainees’ mental wellbeing declined over time, and trainees fear the impacts of a loss of clinical learning on their future job roles in the healthcare workforce. Healthcare employers should be mindful of trainees’ perceived gaps in knowledge and skills and risk for “imposter syndrome”. Organisations should consider providing additional mentoring and support for new recruits to the healthcare workforce. This may help to increase their opportunities to discuss or practice clinical skills and build their confidence with relation to their clinical competencies. Despite challenges with new approaches to teaching delivery, trainees value the efforts made by academic and support staff in ensuring the continuation of theory, practice learning and assessments through a challenging time. Support from academic tutors is highly valued, but the quality of support varies. Over a year into the pandemic, there were evident deficiencies in structured support systems for student mental wellbeing, particularly around awareness of and access to services. Higher education institutions should consider further resource investment in structured support services to expand service capacity.

## Figures and Tables

**Figure 1 ijerph-20-04255-f001:**
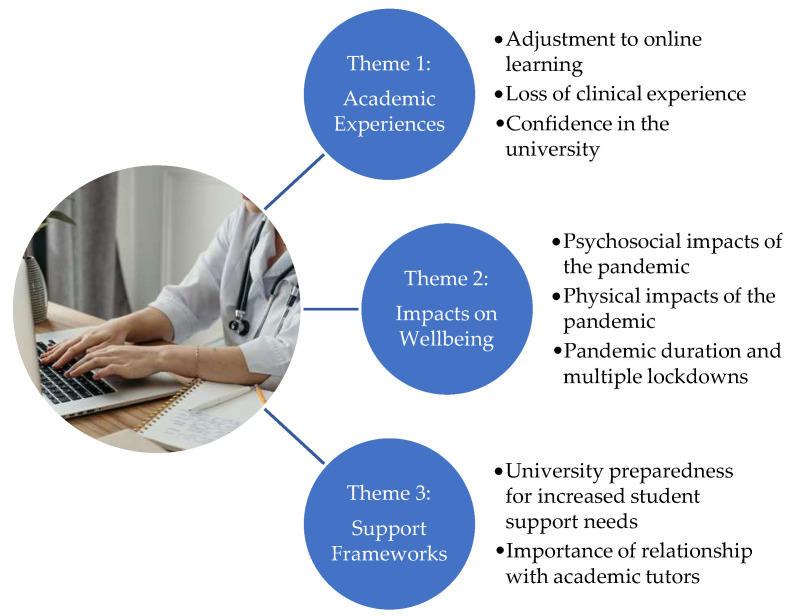
Summary of themes and sub-themes.

**Table 1 ijerph-20-04255-t001:** Participant characteristics.

Participant	Age	Gender	Discipline	Year of Study	COVID-19 High-Risk
1	19	Woman	Medicine	1	No
2	22	Woman	Medicine	4	Yes
3	20	Woman	Medicine	3	Yes
4	20	Woman	Medicine	2	Yes
5	19	Man	Medicine	1	No
6	26	Woman	Medicine	3	Yes
7	42	Woman	Midwifery	3	No
8	21	Man	Medicine	3	Yes
9	21	Woman	Nursing	2	Yes
10	34	Woman	Midwifery	3	Yes
11	18	Woman	Midwifery	1	Yes
12	22	Woman	Nursing	3	Yes

## Data Availability

The data that support the findings of this study are available on reasonable request from the corresponding author. The data are not publicly available due to their containing information that could compromise the privacy of research participants.

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
