# Peer review of "“We’re Not Going to Be as Prepared”: A Qualitative Study of Healthcare Trainees’ Experiences after One Year of the COVID-19 Pandemic"

_ijerph, 2023, doi:10.3390/ijerph20054255_

Round 1

Reviewer 1 Report

The article is well written, relevant and adds knowledge to the area. The objective is clear. The method is well designed. The results are well presented and discussed. The conclusion answers the objective of the study. Therefore, I recommend accepting and publishing the article after some minor grammatical revisions.

1. In the line-94, the author mentioned ‘to agree on relevance of the questioning’

Suggestion: Put article ‘the’ before relevance. i.e., the relevance.

2. In the line-161, it is mentioned ‘Having all the course moved online.

Suggestion: Should be having all the courses moved online.

3. In the line-345, the authors stated that ‘Our trainees felt that live online delivery allow interactivity'.

Suggestion: Please put ‘s’ at the end of word allow.

4. In the line-483, it is stated that ‘COVID-19 pandemic have been recognised’.

Suggestion: It would be ‘has been’ instead of ‘have been’.

Reviewer 2 Report

Does the introduction provide sufficient background and include all relevant references?

Principally, the introduction presents the most essential and indispensable research issues. The whole narrative is conducted in an accessible and correct way, mostly with respective and shrewd references in the footnotes.

Are all the cited references relevant to the research?

The predominant part of references cited in the article have been prepared in an adequate, complete and correct manner. Regrettably one slight error have occured.

Namely, there can be indicated an inadequate use of footnote 15 reference. The reference should regard solely healthcare students ("although the mental health of healthcare trainees has also been impacted during the COVID-19 pandemic [10-16]." - see 33-34), whereas footnote 15 does not directly apply to this group ("As medical and nursing students may be at greater risk for developing mental health issues from working in clinical environments and coming into close proximity with patients (Chandratre, 2020), we did not include studies that specifically sampled medical and nursing students to prevent possible overestimations of the overall prevalence." - see article invoke in footnote 15).

Is the research design appropriate?

The sequence of the presentation of the research acquis proposed in the reviewed article is accurate, as well as compatible with the classical scholar methodology applied to social sciences. All necessary elements were properly highlighted (introduction, methods, results, discussion, and conclusion).

Are the methods adequately described?

The research methods have been presented in detail and comprehensive manner. Particularly noteworthy is fully transparent presentation of the methodology with the usage of the divions on the study design, participants and setting, procedure, and also data analysis.

Are the results clearly presented?

The results of the conducted research were properly presented. Nonetheless a slightly broader scope of the detailed information provided to participants might have seemed justified. Videlicet, it have been mentioned, that study participants gave their "written informed consent" (see 81). Whereas in the content of article there were no futher clarification which specific information were provided to participants.

Are the conclusions supported by the results?

Principally the conclusions are properly justified and supported by the conducted research and its results. Morover, significant strengthening as well as limitating factors of conducted study were correctly or satisfactorily indicated.

Nonetheless, there may be indicated one too far-reaching conclusions. Specifically, it was pointed out that "Healthcare employers should be mindful of potential or perceived gaps in knowledge and skills and consider providing additional mentoring and support for new recruits to the healthcare workforce." (see 499-501).

However, no such conclusions arise from the results of the research conducted. This is because only the students' statements (precisely their opinions and concerns) were pointed out, not the actual assessment of their skills and abilities by those competent to do so (their lecturers and examiners). After all, a possible deficiency of clinical experience in future work would have to result in a failing grade and failure to pass the semester or year (otherwise that would be a breach of educational law regluations).

Interestingly enough, the same view has also been directly provided by in the reviewed article's Results, namely: "Trainees 188 highlighted a need for clinical mentors to recognise the impact of the pandemic on clini-189 cal learning when students transition to professional posts, and a desire for future super-190 visors to recognise that they may need more support than their predecessors" (see 188-191).

Hence, this above-mentioned conclusion (referring healthcare employers) should be cosindered as rather insufficiently justified and thus debatable.

These above-mentioned remarks in turn determine the assesment on Originality / Novelty, Significance of Content, Quality of Presentation, Scientific Soundness, Interest to the readers.

In conclusion, the article may be accepted after minor revision.

Reviewer 3 Report

General comment:
The study was designed to explore the experiences of healthcare trainees after one year of the COVID-19 pandemic using a qualitative approach. This study addresses an important issue that can provide valuable information on the experiences of healthcare trainees and their coping mechanisms post-Covid-19.

Introduction
The introduction section discussed the problem, but the literature review is generally shallow, and the authors have not provided strong arguments why a study of this magnitude is necessary. It is also unclear what gaps this study is filling from previous literature and why such gaps need filling. The work was not founded on any relevant theory.

Methods
Authors should provide the number of participants in the following categories - medicine, nursing and midwifery trainees registered at higher education institutions in the United Kingdom (UK).
Provide more details on how these participants were selected. What were the eligibility and exclusion criteria? In the abstract, the authors stated that 10 females and 2 males participated in the study. Add this information to the methods section. Also, explain why the number of females selected exceeded that of the males by a wide margin.

Results
Satisfactory

Discussion
The discussion is generally well-written using existing literature. However, the authors should discuss the implications of each finding.

Study strengths and limitations
The authors have done well in identifying the strengths and weaknesses of their study. However, suggestions for future research should be made based on the weaknesses of the study.

References
AAuthors should provide the web addresses or digital object identifiers (DOIs) of all referenced materials.
